# Ultrafast photomechanical transduction through thermophoretic implosion

Nikita Kavokine [1,4], Shuangyang Zou [2,4], Ruibin Liu[2], Antoine Niguès [1], Bingsuo Zou [2,3]* & Lydéric Bocquet [1]*

Since the historical experiments of Crookes, the direct manipulation of matter by light has been both a challenge and a source of scientific debate. Here we show that laser illumination allows to displace a vial of nanoparticle solution over centimetre-scale distances. Cantilever-based force measurements show that the movement is due to millisecond-long force spikes, which are synchronised with a sound emission. We observe that the nanoparticles undergo negative thermophoresis, and ultrafast imaging reveals that the force spikes are followed by the explosive growth of a bubble in the solution. We propose a mechanism accounting for the propulsion based on a thermophoretic instability of the nanoparticle cloud, analogous to the Jeans's instability that occurs in gravitational systems. Our experiments demonstrate a new type of laser propulsion and a remarkably violent actuation of soft matter, reminiscent of the strategy used by certain plants to propel their spores.

[1] Laboratoire de Physique de l'École Normale Supérieure, ENS, Université PSL, CNRS, Sorbonne Université, Université Paris-Diderot, Sorbonne Paris Cité, Paris, France. [2] Beijing Key Lab of Nanophotonics and Ultrafine Optoelectronic Systems, Beijing Institute of Technology, Beijing 100081, China. [3] Key Lab of Featured Metal Resources Utilization and Advanced Materials, School of Physics, Guangxi University, Nanning 530004, China. [4] These authors contributed equally: Nikita Kavokine and Shuangyang Zou. *email: zoubs@bit.edu.cn; lyderic.bocquet@ens.fr

In 1874, Crookes[1] observed that a light-absorbing vane placed in a vacuum-filled glass bulb would rotate when exposed to sunlight and interpreted the results as the effect of radiation pressure. Crookes' interpretation was the cause of much debate at the time and it was 5 years later that Maxwell proposed the currently accepted explanation: the vane actually rotates due to thermophoresis of the residual gas molecules in the bulb[2]. Since then, a variety of methods for propelling macroscopic objects with light have been proposed[3], involving, if not radiation pressure[4,5], the light-induced ejection of matter, resulting in propulsion through momentum conservation[6–8]. However, the potential of thermophoresis—the driving mechanism in Crookes' experiment—for macroscopic light-induced propulsion has hardly been explored since the nineteenth century. Although light-induced self-thermophoresis has been highlighted as a means of controlled optical manipulation of individual colloids[9,10], light-induced thermal gradients have been overlooked as a means of macroscopic actuation.

We describe in this study a macroscopic system that is propelled over centimetre-scale distances by the sole action of light, yet without any exchange of matter with the surrounding medium; we show that the propulsion mechanism is based on light-induced thermophoresis.

## Results

**The photomechanical effect.** Our system consists of a closed vial containing 1 mL of a concentrated solution of lead sulphide (PbS) nanoparticles in cyclohexane. The particles have an average diameter of 8 nm and strong absorption in the near-infrared (Supplementary Fig. 1). We observe that when illuminated with a ∼1.5 W, 975 nm laser, the vial suddenly 'jumps' away from the laser source (Fig. 1a and Supplementary Movie 1), over a few millimetres. The typical velocity of the vial during a jump is about 1 cm s$^{-1}$, corresponding to a mechanical work around 30 µJ. After one such jump, the vial falls out of range of the laser, which

diverges from a fibre tip. Moving the fibre tip closer to the vial results in another jump and the process can thus go on, yielding propulsion of the vial over several centimetres. In the experiment shown in Fig. 1, an average speed of 1 mm s$^{-1}$ was obtained (Fig. 1b and c, and Supplementary Movie 2), which was essentially limited by the rate of laser repositioning.

**Macroscopic characterisation.** To characterise this intriguing phenomenon, we built a force-measurement setup based on a cantilever, whose deflection is monitored using a quadrant photodiode (Fig. 2a). The vial was glued on the cantilever and its deflection was recorded as a function of time upon application of the laser light. Then, knowing the cantilever stiffness (Supplementary Fig. 2), we could translate the observed deflection into the horizontal component of the force exerted on the vial-cantilever system. A typical measurement is shown in Fig. 2b: the cantilever registers a series of force spikes, which starts when the laser is switched on; it ceases as soon as it is switched off. Each force spike is accompanied by the emission of audible sound (Fig. 2b) and ∼97% of the recorded sound spikes match a force spike. A typical force spike lasts around 5 ms and contains several oscillations of the force between positive and negative values (Fig. 2b, inset). It is noteworthy that a positive force is oriented here along the direction of propagation of the laser and the initial increase of the force is always towards positive values. Varying the laser power, one observes that the force spikes are triggered only above a threshold power, as shown in Fig. 2c; above the threshold, the average spike frequency (i.e., the number of spikes per unit time) increases with increasing power. The threshold laser power depends on particle concentration: the lower the concentration, the higher the threshold. For a 2% in weight particle concentration, no spikes were observed up to 7 W laser power. The average amplitude of a spike is around 6 mN; remarkably, it does not have any appreciable dependence on laser power or particle concentration (Fig. 2d).

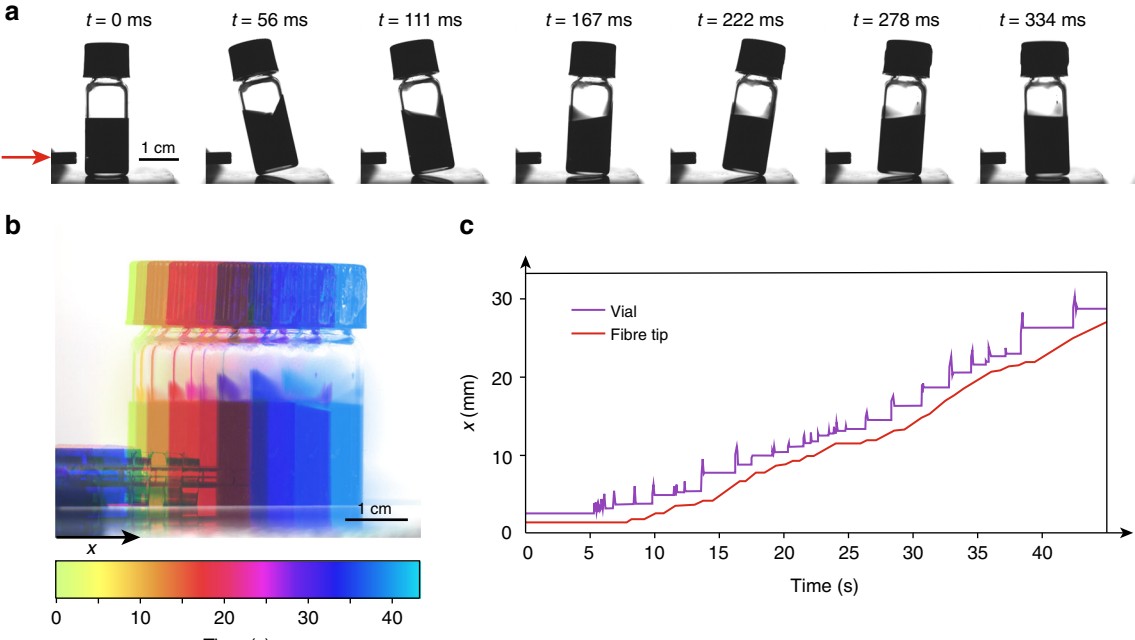

**Fig. 1 A vial filled with a lead sulfide nanoparticle solution is propelled by a near-infrared laser. a** Snapshots of the vial motion during one laser-induced jump. The red arrow indicates the propagation direction of the laser. **b** Temporal colour-code projection of the large-scale vial motion obtained when the fibre tip is kept at an approximately constant distance from the vial. The image is a superposition of ten snapshots of the vial motion, with the colour encoding time. **c** Vial and fibre tip position as a function of time corresponding to the large-scale motion shown in **b**.

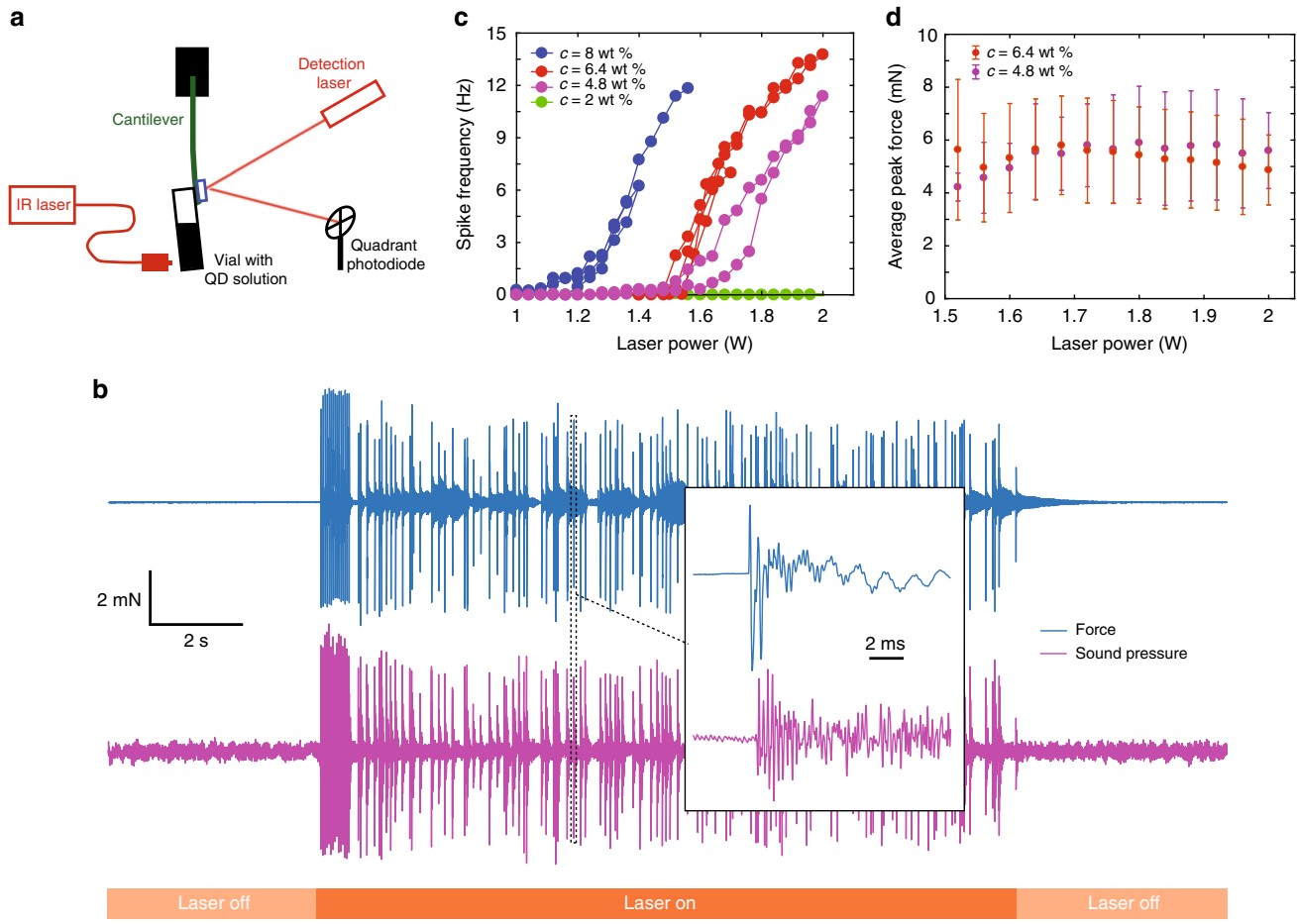

**Fig. 2 Cantilever-based measurements of the laser-induced force. a** Schematic of the experimental setup. The vial is suspended on a cantilever; a low power laser and a quadrant photodiode are used to monitor the cantilever position as a function of time. **b** Typical force and sound pressure vs. time measurement. The particle concentration is 6.4 wt% and the laser power is 1.5 W. Inset: zoom on a single force spike and the corresponding spike in sound pressure. **c** Frequency of the force spikes as a function of laser power, for different particle concentrations. Spikes are observed above a critical laser power, which increases with decreasing concentration. **d** Average peak force as a function of laser power, for two different particle concentrations. Error bars represent the SD.

**Microscopic mechanism.** We investigated the origin of the photomechanical effect in light of the macroscopic characterisation described above. Radiation pressure could clearly be ruled out, as it acts continuously and not in spikes; moreover, one can estimate the radiation pressure in our system at a value around 5 nN, which is orders of magnitude below the forces that are measured, in the several milli-Newton range. Furthermore, the vial is sealed and can hardly be expected to exchange matter with the surrounding medium. Thus, the momentum of the vial and solution it contains must be conserved and if the vial moves away from the laser, then the fluid inside must acquire momentum towards the laser. To understand how the fluid acquires this momentum, we placed the nanoparticle solution in a 1 mm-thick spectroscopic cuvette and synchronised the force measurement with high-speed imaging of the laser-illuminated region. The results are presented in Fig. 3.

As opposed to the large vial, where force spikes could be observed indefinitely (we observed no change in behaviour for up to 30 s), in the thin spectroscopic cuvette the spiking ceased after less than a second of illumination (Fig. 3b). After about 10 s, imaging of the illuminated region revealed the formation of a solid aggregate (Fig. 3c and Supplementary Movies 3 and 4), due to the PbS particles accumulating next to the cuvette wall, somehow 'jamming' the spiking mechanism. Therefore, there is a particle–laser interaction that causes the particles to migrate towards the laser. Such migration could be due to the direct interaction of the particles with the laser electric field: the phenomenon in question would be dielectrophoresis[11]. However, one can estimate the dielectrophoretic migration velocity in our system, as <1 nm h$^{-1}$ (see Supplementary Discussion); hence, the contribution of dielectrophoretic driving is negligible. On the other hand, one could expect the particle migration to be the result of temperature driving, as the particles have strong absorption at the laser wavelength. Indeed, when performing infrared imaging (see Supplementary Movie 5), we observe temperature differences of up to 30 K across the system and the temperature gradient reaches up to 2 K mm$^{-1}$. We expect such temperature differences to drive particle motion through thermophoresis: here, negative thermophoresis, as particles move towards higher temperatures. The particle flux $j(\mathbf{r})$ due to thermophoresis is characterised by the Soret coefficient $S$, defined by:

$$j(\mathbf{r}) = -DS\rho(\mathbf{r})\nabla T(\mathbf{r}), \qquad (1)$$

where $D$ is the particle diffusion coefficient and $\rho(\mathbf{r})$ is the particle concentration[12,13]. Thermal imaging combined with monitoring of the aggregate size as a function of time (Supplementary Fig. 3) allowed us to make a rough estimate of the Soret coefficient of the PbS particles, yielding $S \approx -4$ K$^{-1}$. To our knowledge, there have been no reported measurements of the Soret coefficient for PbS particles in cyclohexane. For a more common experimental system, polystyrene particles in water, Soret coefficient values

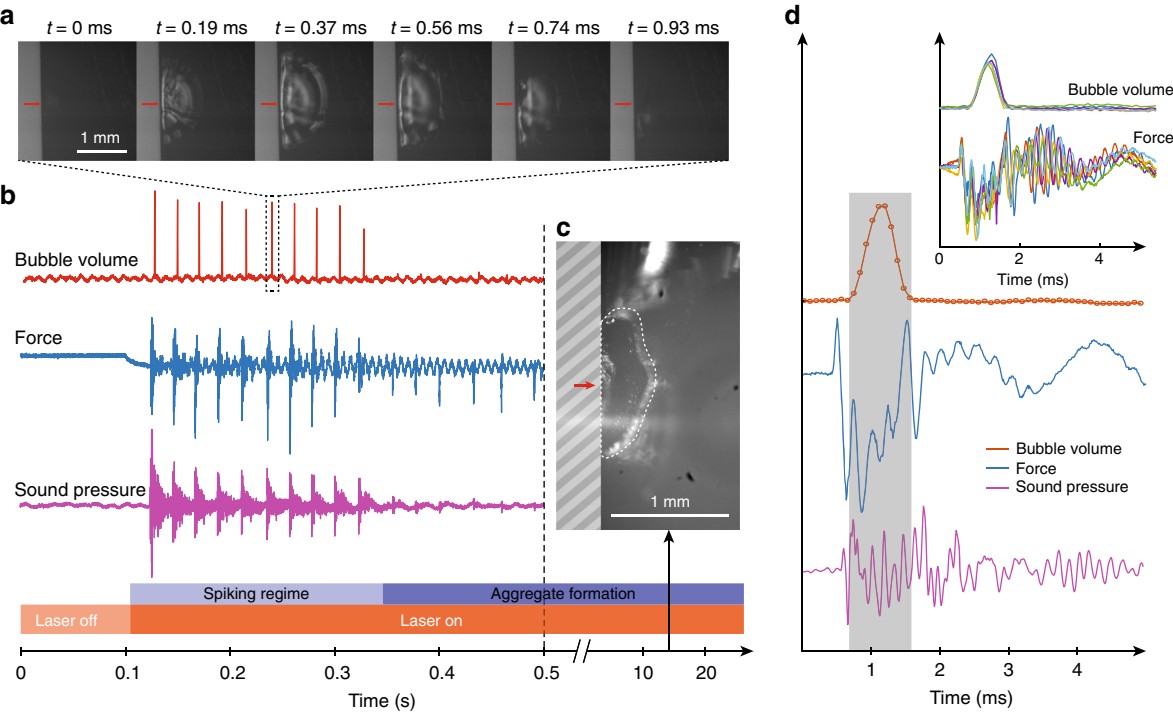

**Fig. 3 Microscopic investigation of the origin of the photomechanical effect.** The four panels show results of simultaneous measurement of force and sound pressure, coupled to ultrafast imaging (11,000 fps), in a 1 mm-thick spectroscopic cuvette. **a** High-speed photography snapshots of the dynamics of a bubble growing and collapsing upon laser illumination. The red arrow indicates the laser impact point. **b** Typical recording of force and sound vs. time and of the volume of the bubble in solution as determined by optical imaging. **c** Image of the particle aggregate that becomes visible after about 10 s illumination. The red arrow shows the impact point of the laser, the grey striped rectangle emphasises the cuvette wall and the dashed white line highlights the aggregate in the image. **d** Bubble volume, force and sound pressure vs. time, averaged over the first six spikes shown in **b**. The grey rectangle is a guide to the eye highlighting that the onset of the force spike precedes bubble growth. Inset: aligned time traces of six force spikes, showing very reproducible synchronisation with the bubble dynamics.

ranging from −1.5 to +40 K⁻¹ have been reported[12–16], depending on conditions and particle size. Our estimate is therefore not unreasonable with regard to this range and corresponds to quite a strong, negative thermophoresis.

Furthermore, high-speed imaging of the solution revealed another striking phenomenon. We observed that every force spike is accompanied by the explosive formation of a bubble next to the cuvette wall (Fig. 3a, Supplementary Movie 3). The bubble grows in about 0.5 ms and collapses as rapidly, which could be a signature of either cavitation[17,18] or explosive boiling[19,20]; in that case, our bubble would be analogous to a plasmonic bubble[21]. One could expect these violent bubble dynamics to be the cause of the observed photomechanical effect. However, careful time resolution of the dynamics contradicts this hypothesis. Indeed, the bubble growth results in fluid moving away from the laser, which should trigger macroscopic motion of the vial towards the laser through momentum conservation. This is in contradiction with the macroscopic observation that the vial always moves away from the laser source. Now, one could argue that the bubble dynamics may still be responsible for the vial motion if the necessary momentum was released during bubble collapse. However, this scenario is again contradicted by the experimental observations, as high-speed imaging shows that the vial takes off while the bubble is still growing (Supplementary Movie 1). A further confirmation of this macroscopic observation is provided by the parallel high-speed imaging and force measurement (Fig. 3d), whose synchronisation we ensured down to one camera frame (90 μs, see Supplementary Fig. 5). The time traces of the various measured quantities were found to be very reproducible over several spikes (see Fig. 3d, inset), in terms of shape, time

delay and order of occurence. Thus, the measurement unambiguously shows that the onset of the force spike precedes bubble collapse and even precedes bubble growth; moreover, the spike contains oscillations on a timescale, which is about three times faster than the bubble dynamics. Hence, the bubble seems to arise as a collateral consequence of another phenomenon that is actually responsible for the photomechanical effect.

**The thermophoretic instability.** Based on these various observations—and the dismissal of several scenarios—we propose a physical origin for the photomechanical effect, which is based on a collapsing instability of the colloidal suspension. The key point in our reasoning is that the thermophoretic force that drives the particles towards the laser also induces their mutual attraction. It was proposed theoretically that such a thermophoretic attraction could lead to an instability, analogous to the Jeans' instability observed in gravitational systems[22,23], as we detail in the following.

Assuming for simplicity that all the particles at positions $\mathbf{r}_j$ are illuminated with intensity $I$, each particle behaves as a point heat source and the temperature field satisfies $\kappa \Delta T(\mathbf{r}) = 4\pi\sigma I \sum_j \delta(\mathbf{r} - \mathbf{r}_j)$, where $\sigma$ is the particle absorption cross-section and $\kappa$ the thermal conductivity of the solvent. This leads immediately to a temperature field which depends on the colloidal structure according to

$$T(\mathbf{r}) = T_0 + \frac{\sigma I}{\kappa}\sum_j \frac{1}{|\mathbf{r} - \mathbf{r}_j|}, \qquad (2)$$

where $T_0$ is the temperature far away from the laser. Now, if the

particles undergo thermophoresis characterised by the Soret coefficient $S$, then the particle flux, given by Eq. (1), can be rewritten introducing the effective potential $\mathcal{V}_{\text{eff}}(\mathbf{r})$ which represents the 'thermophoretic interaction':

$$\mathbf{j}(\mathbf{r}) = -DS\rho(\mathbf{r})\nabla T(\mathbf{r}) \equiv \frac{D}{k_B T_0}\rho(\mathbf{r})(-\nabla\mathcal{V}_{\text{eff}})(\mathbf{r}), \qquad (3)$$

with

$$\mathcal{V}_{\text{eff}}(\mathbf{r}) = -\mathcal{G}\sum_j \frac{1}{|\mathbf{r}-\mathbf{r}_j|}, \qquad (4)$$

and $\mathcal{G} \equiv k_B T_0 |S|\sigma I/\kappa$. Therefore, the laser-induced thermophoresis results in the particles interacting with a $1/r$ attractive potential ($S$ is negative), which is analogous to a gravitational potential, and $\mathcal{G}$ plays the role of a gravitational constant. Now, an ensemble of particles—say a cloud of radius $R$ —with gravitational interactions is known to undergo a Jeans' instability when it exceeds a critical mass[24]: quantitatively, the cloud collapses to a point when the gravitational energy per particle exceeds the thermal energy:

$$\rho R^2 \mathcal{G} \sim k_B T_0. \qquad (5)$$

Such a collapsing instability explains the formation of stars from clouds of interstellar dust[25]. The profound analogy between gravitational and certain types of colloidal interactions was first noted by Keller and Segel[26] in the case of chemotactic bacteria[27]: when bacteria are attracted by a molecule that they themselves produce, they may collapse on each other to form dense aggregates. Similar behaviour was later highlighted with diffusiophoretic [28,29], or even capillary[30] interactions, and more recently predicted theoretically for thermophoretic interactions such as the ones considered here[22,23]. It is therefore likely that the analogue of Jeans' instability with thermophoretic interactions explains the brutal force release that we observe. This is further supported by the scaling law argument we develop in the following.

The dynamics of the thermophoretic collapse couple the particle transport in the pseudo-gravitational field to the fluid dynamics. The latter is described with a Navier–Stokes equation:

$$\mu_s[\partial_t\mathbf{v} + (\mathbf{v}\cdot\nabla)\mathbf{v}] = -\nabla p + \rho(-\nabla\mathcal{V}_{\text{eff}}) + \eta\Delta\mathbf{v}, \qquad (6)$$

where $\mathbf{v}$ is the velocity field, $p$ is the pressure, and $\mu_s$ and $\eta$ the suspension mass density and viscosity, respectively. The driving term $\rho(-\nabla\mathcal{V}_{\text{eff}})$ takes its origin in the thermophoretic interaction introduced above. Solving this equation in the presence of the thermophoretic interaction represents a formidable challenge, but one may propose some scaling relations for the collapse dynamics. First, the observations indicate that the collapse occurs over a short timescale, of the order of 100 µs (Fig. 3b). This suggests that the Reynolds number associated with the collapse is relatively large: indeed, using a millimetric size for the collapsing region, one can estimate $\mathcal{R}e \approx 10$; thus, the dynamics are dominated by the transient (inertial) terms in the Navier–Stokes equation, while viscous terms should be small. As a further note, one may remark that the viscous (shear) term cancels for an incompressible flow with spherical symmetry as expected in the present geometry; hence, discarding viscosity effects on a more general ground. The collapse thus results from the balance between the transient inertial term $\mu_s R^3 \dot{\mathbf{v}} \sim \mu_s R^4/\tau^2$ (integrated over the size $R \sim 1$ mm of the collapsing cloud) and the total thermophoretic force $\sim \rho R^3 (-\nabla\mathcal{V}_{\text{eff}})$, with $\mathcal{V}_{\text{eff}}$ the thermophoretic interaction potential defined in Eq. (3). As the pseudo-gravitational potential results from the sum of the interactions of particles within the sphere of radius $R$, one estimates $\mathcal{V}_{\text{eff}} \approx -\rho R^3 \frac{\mathcal{G}}{R}$. This leads to a scaling law for the timescale $\tau$ of the collapse, as

$$\mu_s \cdot \frac{R^4}{\tau^2} \sim \rho^2 \mathcal{G} R^5 \sim \rho R^2 k_B T_0 \qquad (7)$$

hence,

$$\tau \sim \sqrt{\frac{\mu_s R^2}{\rho k_B T_0}}, \qquad (8)$$

Due to the thermophoretic attraction, the density in the illuminated region is expected to be close to the close packing density of the PbS particles, so that assuming $\rho \sim 10^{24}\,\text{m}^{-3}$, we find $\tau \sim 100$ µs. This estimate is indeed in good agreement with the experimental result and justifies a posteriori the inertial assumptions on the dynamics. As a further prediction, the force generated in the collapse can be estimated as the collapsing mass times its acceleration:

$$F \sim m\rho R^3 \frac{R}{\tau^2} = \frac{m\rho^2}{\mu_s} k_B T_0 R^2. \qquad (9)$$

We find $F \sim 40$ mN, which is again compatible with the experimentally observed range.

## Discussion

The above considerations allow us to propose a complete scenario for the origin of the photomechanical effect, which is summarised in Fig. 4. The PbS particles absorb the laser light and become point heat sources. They undergo negative thermophoresis and thus migrate towards the laser source. The particle density increases in the illuminated region, up to the point where the Jeans' instability occurs, resulting in collective motion of the particles towards the laser, giving the whole vial momentum away from the laser. The 'gravitational' collapse is expected to result in a temperature increase and pressure decrease near the wall of the vial, resulting from the increase in the particle concentration and their rapid motion, respectively. This triggers a cavitation event, with a bubble explosively growing and then collapsing next to the wall. The resulting dynamics disperse the particles that have accumulated near the wall[18,31], so that the process can start again and result in a new propulsion event. We expect that the momentum of the particles is transmitted to the vial wall, resulting in the propagation of a shock wave through the glass, which would be responsible for the oscillations observed in the force spikes and the sound: the observed oscillation frequencies were different in the vial and the spectroscopic cuvette (Supplementary Fig. 4). When the vial stands freely on a substrate, the momentum carried by the shock wave can be transmitted to the substrate and then back to the vial so that it takes off: this momentum transfer is similar to what occurs in the propulsion of the Mexican jumping beans[32]. We thus expect only the first positive force peak to matter: the subsequent oscillations contained in the spike (Fig. 4a) occur when the vial is already in the air and they cannot therefore contribute to propulsion.

We have demonstrated the propulsion of a macroscopic object by the sole action of light through a novel mechanism involving negative thermophoresis as a key ingredient. A 1.5 W laser illumination is sufficient to propel a vial weighing 3.5 g at a 1 mm s$^{-1}$ average speed. Remarkably, it is the interplay between three different phenomena—thermophoretic migration, the Jeans' instability and bubble cavitation—which results in the system propelling itself through repeatable, discrete force spikes. We believe it is also remarkable that a 'soft' system, whose dynamics are usually viscous, actually exhibits ultrafast transport, way beyond the viscous timescale. The slow energy accumulation followed by a fast release is reminiscent of the process used by certain plants, such as the fern, to propel their spores[33]. Our novel

**a**

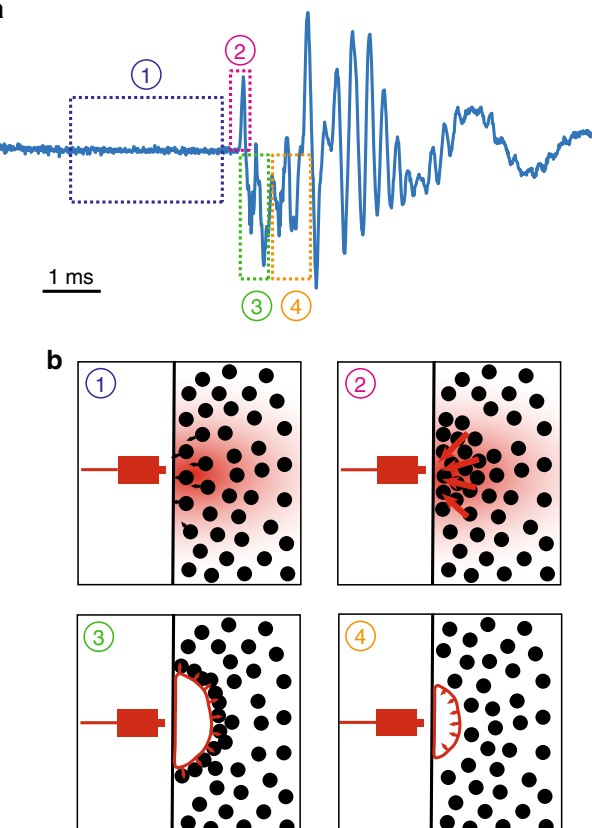

**b**

**Fig. 4 Schematic mechanism of the photomechanical effect. a** Force vs. time signal corresponding to a force spike. The four dashed rectangles correspond to the four steps in the proposed mechanism for the generation of a spike. **b** Schematics for the four steps highlighted in **a**. The black circles correspond to PbS particles and the red colour represents a temperature gradient. (1) Phoretic motion of the particles towards the laser source. (2) Jeans' instability, resulting in brutal force release. (3) Explosive growth of a bubble, which disperses the accumulated particles. (4) Collapse of the bubble and return to the initial state.

mechanism for soft matter actuation could therefore be of interest for mimicking certain bio-inspired functionalities.

## Methods

**Nanoparticle synthesis and characterisation**. The PbS nanoparticles used in all experiments were synthesised following a typical hot-injection method, involving injection of a sulfur precursor into a lead precursor in an organic solvent. All the syntheses were carried out under air-free conditions using a standard Schlenk-line setup. PbS quantum dots purchased from Merck exhibited the same behaviour as the in-house-synthesised ones.

Transmission electron microscopy (JEOL-2100F) was used to visualise the nanoparticles. Their crystal structure was obtained by the X-ray diffraction (D8-Advance X-ray diffractometer). The absorption spectra in cyclohexane were recorded using a Shimadzu UV-3600 spectrometer.

**Force-measurement setup**. A schematic of our force-measurement setup is given in Fig. 2a. A Sheaumann HF-975-7500-25C fibre-coupled laser diode, powered by an Arroyo LaserSource controller, was used for actuating the system. The vial or spectroscopic cuvette were glued to a 3 cm-long metal cantilever (made from a metal ruler). A small mirror was glued to the other side of the cantilever. A low power red laser was reflected on the small mirror and directed onto a quadrant photodiode (Thorlabs, PDQ80A). Laser illumination above the threshold power (see Fig. 2c) resulted in force spikes, which induced deflection of the cantilever in the 10 μm range. The resolution in cantilever deflection was below 100 nm and the time resolution was given by the photodiode bandwidth (150 kHz). In all experiments, the excitation laser fibre tip was placed at exactly 1 mm from the vial wall using a stepper motor and 2 mm above the vial bottom. The analogue control of the laser power, camera triggers, and force and sound measurements were

synchronised using a National Instruments DAQ card (NI USB-6363) and a custom Labview software.

**Imaging**. High-speed imaging was performed using a Phantom v642 or a Hamamatsu Orca Flash 4.0 camera and either a Nikon 50 mm macro lens or a Thorlabs MVL12X3Z 12× zoom lens, with coaxial illumination from an Olympus U-HGLGPS (130 W) light source. Thermal imaging was performed using a FLIR Q655sc infrared camera.

## Data availability

All data associated with the plots is available from corresponding author upon reasonable request.

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

## Acknowledgements

We thank A. Kavokin for initiating the collaboration, and L. Canale, A. Lainé, T. Mouterde, P. Robin, D. Baigl and D. Lohse for discussions. We are indebted to J. Delannoy, A. Bouillant, C. Cohen and D. Quéré for lending the high-speed and infrared cameras. L.B. acknowledges funding from the EU H2020 Framework Programme/ERC Advanced Grant agreement number 785911-Shadoks. The work was partially funded by the Horizon 2020 programme through 766972-FET-OPEN-NANOPHLOW and by Agence Nationale de la Recherche (France) through the project ILLIAD.

## Author contributions

R.L. observed the photomechanical effect. S.Z. synthesised the nanoparticles and performed initial experiments with supervision from B.Z. N.K. performed all experiments reported in the paper with contributions from A.N. N.K. and L.B. analysed the data and wrote the paper. All authors discussed the results and commented on the manuscript. B.Z. and L.B. supervised all aspects of the project.

## Competing interests

The authors declare no competing interests.
