## [Peer Review File · Nature Communications]

Reviewers' comments:

Reviewer #2 (Remarks to the Author):

The paper reports on the striking observation of photomechanical transduction. It reports both very carefully done experiments and a theoretical explanation. Clearly, the phenomenon is general interest and this excellent and original work should be published in a high-profile journal. But the authors first may want to consider the following comments:

1. Force and sound in figure 2b seem to be correlated, but to see this better, an extra zoom-in is needed. Also figure 3b is not sufficiently zoomed in. The authors should also provide the correlation coefficient, and best how it depends on the control parameters.
2. The sound emission from a collapsing spherical bubble is proportional to $(2 R \dot{R}^2 + R^2 \ddot{R})$, according to the Rayleigh-Plesset dynamics. Do the authors' volume measurement $Vol \sim R^3$ allow for an (upper) estimate of this sound emission? Would be very useful. In particular for the correlation. Cf. figure 3d.
3. Comparison of $R(t)$ or $Vol(t)$ as measured with the Rayleigh-Plesset dynamics would be useful.
4. Am not totally sure that the observed bubble is not a (giant) plasmonic bubble, see Y. Wang et al., PNAS 115, 7676 (2018). This would not account for the propulsion of course. I am convinced by the authors' model.
5. In this context it would be good to know the size of the nanoparticles. I did not see it given in the text.
6. Figure 4b-4 and explanation: I do not think that the particles will be homogeneously distributed after the event. Inhomogeneities will remain.

Minor:

- Page 3, 2nd line: Fig 2d, not fig 1d.

Reviewer #3 (Remarks to the Author):

The authors describe a very interesting photomechanical phenomenon observed for a suspension exposed to light. Macroscopic motions are produced and traced to small scale colloidal responses to the particles absorbing laser light. The macroscopic characterization is well documented in the paper. Certainly the macroscopic phenomenon is fascinating, as are the small scale measurements. I tried several times to understand the mechanism proposed by the authors but in the end failed to understand what they were proposing, i.e. I can repeat what they write but I was not left convinced the underlying effect was thermophoresis of the particles, in part because they seemed to indicate that for similar materials (though not for polystyrene) their thermophoretic estimate was two order of magnitude too high. I can imagine the paper being very interesting to readers, at least given the macroscopic characterization, but I find the current description of the "explosive dynamics" unclear – I would recommend a revision to try to clear this up further. Also, have the authors at all considered that particles could be attracted to the laser by a dielectrophoretic mechanism? So I find the paper very interesting but I admit to being puzzled by what is written about the colloidal details and wonder whether any other readers will understand the mechanism proposed.

Additional remarks and questions:

- 1) p. 2: "containing 1 mL of a concentrated solution of lead sulphide nanoparticles in cyclohexane"
– What is the concentration or volume fraction of nanoparticles? What is the diameter of the

particles?

2) p. 2: "the laser, which diverges from a fibre tip" – What is the approximate radius of the laser beam in the liquid bath? Is it possible that particles are attracted to the laser by a dielectrophoretic response?

3) p. 2: "In the experiment shown in figure 1, an average speed of $1 \text{ mm} \cdot \text{s}^{-1}$ was obtained" – Since the authors wrote that they have to re-position the laser how does this statement about average speed have any meaning unless they give more details, e.g. the rate of re-positioning.

4) p. 2-3: "The average amplitude of the force spike is independent of the particle concentration or laser power." Shouldn't the concentration affect the time between the spikes? Has this been characterized (or was it in the paper and I missed it)?

5) p. 5: "We thus expect that the observed accumulation of particles results from the temperature gradient that the particles themselves create, meaning that they undergo negative thermophoresis." – This is unclear to me. How can the authors rule out other forces associated with particles focusing toward the laser (e.g. a dielectrophoretic force), which is also consistent with the particles interacting with the light?

In particular, the authors suggest a thermophoretic Soret coefficient 100 times larger than common values (for what materials)? Lead sulphide seems very different to me than polystyrene so I am skeptical about the authors analogy. Are they also skeptical?

6) Figure 1a: indicate the direction of the laser beam. Also, label the direction x in figure 1 (it is used in figure 1c).

7) p. 3: "Fig. 1d" should be Fig. 2d.

8) I do not understand Fig. 2c – the frequencies are a few to 10 Hz but the data in figure 2b seem to be to be much higher frequency. What am I missing? Maybe a factor of 1000 or something of that order is missing.

9) p. 3, last paragraph – the first sentence is a run-on; it needs re-wording.

10) p. 4: "The PbS particles have strong absorption at the 975 nm laser wavelength" – This fact should be stated much earlier when the lead sulfide solution is first introduced.

11) The description in the paper reminded me a little of the 2017 reference: "Laser streaming: Turning a laser beam into a flow of liquid" by Wang et al. Science Advances 2017 Is there anything about the physics that might be related between these works?

12) In the end the authors rule out bubble as the origin of the force, but: "onset of the force spike precedes bubble collapse, and even precedes bubble growth; moreover, the spike contains oscillations on a timescale which is about three times faster than the bubble dynamics". Nevertheless, is it possible that bubble growth (not collapse) in the neighborhood of the wall breaks the symmetry and transfers momentum to the wall? The wall would push back on the fluid and so move the vial in the direction of laser. But the observation quoted above seems to remove the bubble from the force-generating phenomenon. I just was not clear on this since they seem to show that bubble collapse was not relevant – however, I am asking about bubble growth?

13) p. 6: "scenario" – I think "senarios" is more common (except for maybe those well informed about Latin languages).

14) P 7: "thermophoretic collapsing instability of the colloidal suspension. Temperature drivings are indeed at the core of the phenomenon under scrutiny: the PbS particles do absorb light and undergo (particularly strong) negative thermophoresis;" - As noted above, I am not convinced that thermophoresis has been demonstrated so I think their wording needs to be modified since perhaps their statement is more at the level of a hypothesis.

15) p. 7: For the $1/r$ temperature fields and the suggested of an associated thermophoresis, an interesting analogy to a thermal clustering is discussed though I am not familiar with the colloidal research that is cited. I think some clarification/explanation here would help.

16) Equation (5) – G should be defined. Do they mean the gravitational constant?

17) After equation (5): "This relation predicts incidentally that the instability occurs at higher power for lower concentration, in line with our observations" – how is this statement deduced from (5) which as written does not involve concentration or laser power?

18) p. 7-8: The "collapse dynamics" being referred to is about the particle aggregates that form?

19) p. 8: The authors discuss the collapse of the cloud. But aren't the small particles being displaced through a viscous fluid? Isn't that motion occurring at a much smaller Reynolds number so might be viscously controlling the phenomenon? Is this the estimate they are proposing for the Soret coefficient?

20) Equation (6): μ_s is an odd choice for a density.

21) I am puzzling over equation (9) – these estimates are based on a spherically symmetric collapse but then they are applied to the "explain" the order of magnitude of the force measured in the direction of motion. This seems inconsistent to me.

22) Did the authors ever explain the "Jeans instability"? I do not understand the analogy they are trying to make and possibly many readers will also not know the physics analogy being made?

Experiencing the Force: ultrafast photomechanical transduction through thermophoretic implosion

– answer to referees

N. Kavokine, S. Zou, R. Liu, A. Niguès, B. Zou and L. Bocquet

*Unless specifically mentioned, citations point to references provided at the end of this answer. We highlight in **magenta** in our comments the modifications made to the manuscript, and in **blue** the relevant reviewers' comments. In the revised manuscript, changes are highlighted in **red**.*

General comments

We thank both referees for their thorough reading of the manuscript and their positive comments on the paper. In particular, we are glad that the reviewers have agreed on the general interest of the phenomenon we report, and have both highlighted the quality of our experimental work.

We believe we have been able to fully address the concerns expressed by the referees in the point by point responses below, and we have taken their advice into account in the revised manuscript. Overall, in our opinion, the revision has allowed to significantly strengthen the paper.

Answer to reviewer #1

We thank the referee for their very positive feedback on our manuscript. We thank them in particular for acknowledging "very carefully done experiments".

- (1) *Force and sound in figure 2b seem to be correlated, but to see this better, an extra zoom-in is needed. Also figure 3b is not sufficiently zoomed in. The authors should also provide the correlation coefficient, and best how it depends on the control parameters.*

Indeed, there is nearly perfect correlation between force and sound. In the data of figure 2b, one can count 130 force spikes and 134 sound spikes; the 4 extra sound spikes are very small and most probably due to ambient noise. In terms of percentage, for 97% of sound spikes there is a force spike.

In order to define a correlation in a more systematic manner, we first apply a low pass filter, so as to retain only the envelopes of the spikes and not the oscillations inside. Then, we threshold each signal at 10% of its maximum value, and define the correlation as the fraction of time where the thresholded force and sound signals have the same value. The correlation defined in this way is 97%, yielding the same result as simple spike counting.

Hence, following the referee's suggestion, we have added the corresponding sound trace to the inset of figure 2b, and added a note on the correlation between force and sound in the text: "Each force spike is accompanied by the emission of audible sound (Fig. 2b), and 97% of the recorded sound spikes match a force spike.". Figure 3d is a zoom in of figure 3b; we have made this clearer in the corresponding legend: "**d.** Bubble volume, force, and sound pressure versus time, averaged over the first six spikes shown in panel b."

- (2) *The sound emission from a collapsing spherical bubble is proportional to $(2R\dot{R}^2 + R^2\ddot{R})$, according to the Rayleigh-Plesset dynamics. Do the authors' volume measurement $Vol \sim R^3$ allow for an (upper) estimate of this sound emission? Would be very useful. In particular for the correlation. Cf. figure 3d.*

We thank the referee for this interesting suggestion. However, if the sound emission was due to the bubble dynamics, it would occur during the collapse of the cavitating bubble, similarly to what occurs, for instance, when recording sound from a snapping shrimp [1]. However, the data in figure 3d shows that sound emission precedes bubble collapse and even precedes bubble growth, hence it is not due to the collapse of the bubble, but rather the thermophoretic collapse phenomenon that precedes bubble formation (the collapsing particles would induce a shock wave in the vial wall). Therefore, unfortunately, we cannot compare here the sound volume to the Rayleigh-Plesset dynamics.

- (3) *Comparison of $R(t)$ or $Vol(t)$ as measured with the Rayleigh-Plesset dynamics would be useful.*

We thank the referee for this interesting suggestion. The collapse time τ of a bubble of radius R_0 in a fluid of density ρ at pressure P_0 is, according to Rayleigh-Plesset dynamics, $\tau \approx 0.915 \times R_0 \sqrt{\rho/P_0}$. Our bubbles are of radius approximately 1.5 mm, thus taking for ρ the density of cyclohexane yields $\tau \approx 100 \mu\text{s}$, while experimentally we measure $\tau \sim 400 \mu\text{s}$. Thus the order of magnitude predicted by Rayleigh-Plesset dynamics is correct, but only with a fair agreement. This is expected since our experimental situation is very different from a collapsing spherical bubble. The discrepancy may have two sources: (i) the above formula assumes a

spherically symmetric collapse, while in our case the bubble dynamics are hindered by three walls; and (ii) the fluid density around the collapsing bubble is expected to be greater than that of cyclohexane due to the accumulation of particles. As an extreme, if one takes for ρ the close-packed density of the PbS particles one finds $\tau \sim 300 \mu\text{s}$. Thus, the order of magnitude match the observation but a quantitative comparison with Rayleigh-Plesset dynamics remains difficult (a Rayleigh-Plesset-type equation corresponding to our experimental situation would be difficult to solve).

- (4) *Am not totally sure that the observed bubble is not a (giant) plasmonic bubble, see Y. Wang et al., PNAS 115, 7676 (2018). This would not account for the propulsion of course. I am convinced by the authors' model.*

Indeed, the bubble we observe could either be a cavitation bubble or a plasmonic bubble: the sudden rupture of the liquid could be caused either by a local drop in pressure due to very fast flow, or by a local increase in temperature due to the light absorption by the nanoparticles, and our experiment is not able to discriminate between the two. We refer to the latter mechanism as "explosive boiling". In the revised manuscript, we have added the term "plasmonic bubble" as it might indeed be more familiar to readers, **and the reference to the PNAS paper**: *"The bubble grows in about 0.5 ms and collapses as rapidly, which could be a signature of either cavitation [2, 3] or explosive boiling [4, 5]; in that case our bubble would be analogous to a plasmonic bubble [6]"*. We are glad that apart from this discussion of the origin of the bubble, the referee is convinced by the model we propose.

- (5) *In this context it would be good to know the size of the nanoparticles. I did not see it given in the text.*

The nanoparticles have an average diameter of 8 nm. Supplementary figure S1 shows a TEM image of the particles and their size distribution. We have now added the mention of the particle size to the main text: *"The particles have an average diameter of 8 nm and strong absorption in the near-infrared (supplementary figure 1)"*.

- (6) *Figure 4b-4 and explanation: I do not think that the particles will be homogeneously distributed after the event. Inhomogeneities will remain.*

Indeed, we can hardly expect the distribution of particles to be perfectly homogeneous after the bubble collapse. However, it is not a claim we make; also figure 4b does not exhibit something perfectly homogeneous...

- (7) *- Page 3, 2nd line: Fig 2d, not fig 1d.*

We thank the referee for their careful reading, this has been corrected.

Answer to reviewer #2

(0) *The authors describe a very interesting photomechanical phenomenon observed for a suspension exposed to light. Macroscopic motions are produced and traced to small scale colloidal responses to the particles absorbing laser light. The macroscopic characterization is well documented in the paper. Certainly the macroscopic phenomenon is fascinating, as are the small scale measurements. I tried several times to understand the mechanism proposed by the authors but in the end failed to understand what they were proposing, i.e. I can repeat what they write but I was not left convinced the underlying effect was thermophoresis of the particles, in part because they seemed to indicate that for similar materials (though not for polystyrene) their thermophoretic estimate was two order of magnitude too high. I can imagine the paper being very interesting to readers, at least given the macroscopic characterization, but I find the current description of the "explosive dynamics" unclear – I would recommend a revision to try to clear this up further. Also, have the authors at all considered that particles could be attracted to the laser by a dielectrophoretic mechanism? So I find the paper very interesting but I admit to being puzzled by what is written about the colloidal details and wonder whether any other readers will understand the mechanism proposed.*

We thank the reviewer for their careful reading and positive comments on the manuscript. We are glad in particular the reviewer finds the phenomenon we are reporting of general interest, and acknowledges "well documented" characterisation.

However, the reviewer points out that our exposition of the proposed microscopic mechanism lacks clarity. We indeed agree that some explanations in the previous manuscript might be difficult to follow, in particular because they invoke various physical analogies originating from other fields. Thus, following the referee's suggestion, **we have thoroughly revised our description of the "explosive dynamics"**; we believe this has allowed to significantly clear up the manuscript.

The reviewer also (rightfully) questions whether the underlying phenomenon is necessarily thermophoresis, in light of two issues: (i) the agreement of our estimate for the Soret coefficient with literature values and (ii) the possibility that the particles might be attracted by the laser through a dielectrophoretic mechanism. We agree with the reviewer in that some of our statements should be **reworded at the level of the hypothesis: we have done so in the revised manuscript**. However, we have carefully explored the thermophoretic mechanism and we are convinced it provides a solid and consistent explanation, as supported with the arguments that follow.

Actually, we have considered a range of possible mechanisms for the particle dynamics and dismissed them until only one remained. In line with the referee's suggestion, we have indeed envisioned dielectrophoresis in the electric field of the laser as a possible source for the phenomenon at stake. The dielectrophoretic force on a spherical particle of radius R_0 is [7]

$$F(r) = 2\pi R_0^3 \epsilon_s \frac{\epsilon_p - \epsilon_s}{\epsilon_p + 2\epsilon_s} \nabla(\epsilon_0 E(r)^2), \quad (1)$$

where ϵ_s and ϵ_p are the relative permittivities of the solvent and of the particle, respectively, considered here real since both materials in question are insulating; E is the electric field. Now the electromagnetic energy density corresponding to the diverging laser beam is $\epsilon_0 E(r)^2 \approx P/(2\pi r^2 c)$, where $P = 1.5$ W is the laser power. This yields, at a distance $r = 1$ mm from the fibre tip, $F \sim 10^{-23}$ N, which would lead to particle migration at a velocity $v \sim$

$10^{-13} \text{ m} \cdot \text{s}^{-1}$, that is less than 1 nm per hour: this is incompatible with the experimentally observed timescales. **We have mentioned dielectrophoresis in the revised manuscript and we have added the above discussion to the supplementary information.**

Thus, we show that dielectrophoretic driving in our system is expected to be negligible, since the electromagnetic energy gradient is way too small. Temperature drivings, on the other hand, are clearly present, since we directly observe through infrared imaging (see movie S5 and supplementary figure S3) **temperature differences of up to 30 K** across the system; the maximal temperature gradient we measure is $2 \text{ K} \cdot \text{mm}^{-1}$. Hence our statement "temperature drivings are at the core of the phenomenon under scrutiny": it is reasonable to expect that if such huge temperature gradients are present they are going to drive the system in some way.

The driving of particles by a temperature gradient is by definition thermophoresis. Simultaneous imaging of the particle aggregate formation and temperature measurement allowed us to extract a rough estimate of the corresponding Soret coefficient: $S \sim -4 \text{ K}^{-1}$. The referee raised the question of the comparison of this Soret coefficient with literature values. In the literature, to our knowledge, most investigated systems are polystyrene particles in water or various polymers in organic solvents. Based on the two review articles [8, 9], Soret coefficients in these systems were reported in a broad range, from -1.5 to 40 K^{-1} . Both extreme values were reported by Piazza and coworkers in [10] for surfactant coated polystyrene particles; the whole range of Soret coefficients was spanned when varying the particle size and the temperature; similar variation ranges were found by Duhr and Braun [11] and by Putnam et al. [12]. Such a dispersion in the reported values echoes the fact that Soret coefficients have in general a very subtle dependence on the surface state of particles, and actually theoretical models fail to predict Soret coefficient magnitudes and even their sign.

The Soret coefficient in our experiment ($S \sim -4 \text{ K}^{-1}$) is therefore in the range of other reported values in the literature (keeping in mind that we are merely interested here in an order of magnitude of S). Also, there are to our knowledge no reported direct measurements of the Soret coefficient for the type of nanoparticles studied here (lead sulphide nanoparticles, 8nm in diameter, in cyclohexane). Such measurements are actually very challenging given the nanometric size and strong optical absorption of the particles.

Given all the above, it seems very likely to us that our system is driven by thermophoresis. However, since this raised concerns from the referee, **we cleared up our discussion of the thermophoretic mechanism in the revised manuscript.** We also give more details in the point by point responses below.

- (1) *p. 2: "containing 1 mL of a concentrated solution of lead sulphide nanoparticles in cyclohexane" – What is the concentration or volume fraction of nanoparticles? What is the diameter of the particles?*

The particles have 8 nm diameter on average: supplementary figure S1 shows a TEM image of the particles and their measured size distribution. We have now added the mention of the particle size to the main text: **"The particles have an average diameter of 8 nm and strong absorption in the near-infrared (supplementary figure S1)".**

- (2) *p. 2: "the laser, which diverges from a fibre tip" – What is the approximate radius of the laser beam in the liquid bath? Is it possible that particles are attracted to the laser by a dielectrophoretic response?*

The laser beam is approximately 1 mm in diameter when it reaches the liquid bath. In line with our answer to point (0), we have indeed estimated above the resulting dielectrophoretic

force, which turns out to have a negligible contribution to the system dynamics.

- (3) *p. 2: "In the experiment shown in figure 1, an average speed of $1 \text{ mm} \cdot \text{s}^{-1}$ was obtained" – Since the authors wrote that they have to re-position the laser how does this statement about average speed have any meaning unless they give more details, e.g. the rate of re-positioning.*

We thank the referee for the very relevant remark. The propulsion speed was essentially limited by the rate of laser repositioning. Attempting to reposition the laser faster would result in the vial not having time to stabilise between jumps and falling over. We have clarified this in the revised manuscript: "an average speed of $1 \text{ mm} \cdot \text{s}^{-1}$ was obtained (figs. 1b and c, movie S2), which was essentially limited by the rate of laser repositioning".

- (4) *p. 2-3: "The average amplitude of the force spike is independent of the particle concentration or laser power." Shouldn't the concentration affect the time between the spikes? Has this been characterized (or was it in the paper and I missed it)?*

We believe the reviewer might have missed figure 2c, which shows the average spike frequency, that is the inverse of the time between spikes, as a function of laser power for different concentrations. For a given laser power, the spike frequency decreases with decreasing particle concentration.

- (5) *p. 5: "We thus expect that the observed accumulation of particles results from the temperature gradient that the particles themselves create, meaning that they undergo negative thermophoresis." – This is unclear to me. How can the authors rule out other forces associated with particles focusing toward the laser (e.g. a dielectrophoretic force), which is also consistent with the particles interacting with the light? In particular, the authors suggest a thermophoretic Soret coefficient 100 times larger than common values (for what materials)? Lead sulphide seems very different to me than polystyrene so I am skeptical about the authors analogy. Are they also skeptical?*

We thank the referee for pointing out these two relevant issues. We have discussed them above when answering to point 0. We make an estimate of the dielectrophoretic force, which turns out to be negligible, and we discuss literature Soret coefficient values. In fact, our statement about the literature Soret coefficients was misleading, and we have clarified it in the revised manuscript: "To our knowledge there have been no reported measurements of the Soret coefficient for PbS particles in cyclohexane. For a more common experimental system, polystyrene particles in water, Soret coefficient values ranging from -0.5 to $+40 \text{ K}^{-1}$ have been reported [8, 9, 11, 13], depending on conditions and particle size. Our estimate is therefore not unreasonable with regard to this range, and corresponds to quite strong negative thermophoresis."

- (6) *6) Figure 1a: indicate the direction of the laser beam. Also, label the direction x in figure 1 (it is used in figure 1c).*

This has been done in the revised manuscript.

- (7) *p. 3: "Fig. 1d" should be Fig. 2d.*

We thank the referee for their careful reading; this has been corrected.

- (8) *I do not understand Fig. 2c – the frequencies are a few to 10 Hz but the data in figure 2b seem to be to be much higher frequency. What am I missing? Maybe a factor of 1000 or something of that order is missing.*

We believe the referee's remark might arise from a confusion between what we call spike frequency in the paper, that is the number of spikes per unit time, and the frequency of oscillations *within* a spike. We have clarified what we call spike frequency in the revised manuscript: *"the average spike frequency (that is the number of spikes per unit time) increases with increasing power"*.

- (9) *p. 3, last paragraph – the first sentence is a run-on; it needs re-wording.*

We have reworded: *"We investigated the origin of the photomechanical effect in light of the macroscopic characterisation described above."*

- (10) *p. 4: "The PbS particles have strong absorption at the 975 nm laser wavelength" – This fact should be stated much earlier when the lead sulfide solution is first introduced.*

We agree with the referee; we have modified the manuscript accordingly: *"Our system consists of a closed vial containing 1 mL of a concentrated solution of lead sulphide nanoparticles in cyclohexane. The particles have an average diameter of 8 nm and strong absorption in the near-infrared (supplementary figure S1)" (first paragraph after the introduction).*

- (11) *The description in the paper reminded me a little of the 2017 reference: "Laser streaming: Turning a laser beam into a flow of liquid" by Wang et al. Science Advances 2017 Is there anything about the physics that might be related between these works?*

The paper mentioned by the referee reports on the generation of water flows by light thanks to a gold-nanoparticle-decorated cavity and a photoacoustic effect: the whole phenomenon is called laser streaming. The effect under scrutiny is quite different from ours, since the observed phenomenon is a fluid jet rather than propulsion of a macroscopic object. However, since our PbS particles are strongly absorbing, one might expect them to give rise to laser streaming similarly to the gold nanoparticles. This would result in a flow away from the laser, which is the contrary of what we observe in the experiment. Therefore, we believe the similarity between our work and the one by Wang et al. is in that both involve a fluidic phenomenon related to particles heating up in solution, but the physical consequences are very different in the two cases.

Figure 1: Average force and bubble volume time traces as taken from figure 3d, with the black dashed line highlighting the contribution of the bubble dynamics to the force.

- (12) *In the end the authors rule out bubble as the origin of the force, but: "onset of the force spike precedes bubble collapse, and even precedes bubble growth; moreover, the spike contains oscillations on a timescale which is about three times faster than the bubble dynamics". Nevertheless, is it possible that bubble growth (not collapse) in the neighborhood of the wall breaks the symmetry and transfers momentum to the wall? The wall would push back on the fluid and so move the vial in the direction of laser. But the observation quoted above seems to remove the bubble from the force-generating phenomenon. I just was not clear on this since they seem to show that bubble collapse was not relevant – however, I am asking about bubble growth?*

We thank the referee for the interesting question. Indeed, the discussion of fluid momentum during bubble dynamics is tricky. However, the force measurements unambiguously tell us in which direction the momentum goes. The force spikes that are shown in figure 3d contain oscillations at a frequency around 3 kHz, but they also reproducibly show an inflexion towards negative values on the same timescale as the bubble dynamics. The force goes *down* during bubble growth and *up* during bubble collapse, which means that we see experimentally the bubble growth generating propulsion towards the laser. Therefore the bubble growth cannot be responsible for propulsion away from the laser. In fact, in the macroscopic propulsion experiment the momentum generated by the bubble dynamics does not play a role as the bubble grows while the vial is already in the air. Figure 1 of this answer illustrates the argument above.

- (13) *p. 6: "scenario" – I think "senarios" is more common (except for maybe those well informed about Latin languages).*

We have changed said plural to the version preferred by the referee.

- (14) *P 7: "thermophoretic collapsing instability of the colloidal suspension. Temperature drivings are indeed at the core of the phenomenon under scrutiny: the PbS particles do absorb light and undergo (particularly strong) negative thermophoresis;" – As noted above, I am not convinced that thermophoresis has been demonstrated so I think their wording needs to be modified since perhaps their statement is more at the level of a hypothesis.*

We agree with the referee that the mechanism we propose is only a reasonable hypothesis, and we have taken care to make this clear in the wording in the revised manuscript. However, for the reasons detailed above (answer to point 0), we are certain that the particles do undergo

thermophoresis. The referee's doubts are based on the possibility of an alternative (diffusiophoretic) mechanism of particle interaction with the laser, and on our misleading statement on comparison with literature Soret coefficient values. We have reworded this statement in the revised manuscript as explained above, and we have shown that dielectrophoresis yields a negligible contribution, therefore thermophoresis remains the only possible mechanism of particle motion.

- (15) *p. 7: For the $1/r$ temperature fields and the suggested of an associated thermophoresis, an interesting analogy to a thermal clustering is discussed though I am not familiar with the colloidal research that is cited. I think some clarification/explanation here would help.*

We thank the referee for their interest in this physical analogy, which is, we agree, not of general knowledge. **We have added an explanation on this topic in the revised manuscript.**

- (16) *Equation (5) – G should be defined. Do they mean the gravitational constant?*

G is in fact defined below equation (4), but we mistakenly used a different font for the letter G in equation (5). We thank the referee for pointing out this inconsistency.

- (17) *After equation (5): "This relation predicts incidentally that the instability occurs at higher power for lower concentration, in line with our observations" – how is this statement deduced from (5) which as written does not involve concentration or laser power?*

Indeed, this statement might be misleading and it does not add much to the argument: **we have removed it from the revised version.**

- (18) *p. 7-8: The "collapse dynamics" being referred to is about the particle aggregates that form?*

The "collapse dynamics" refers to the microscopic phenomenon underlying the observed force spikes, and not to the formation of the macroscopic aggregate at long times.

- (19) *p. 8: The authors discuss the collapse of the cloud. But aren't the small particles being displaced through a viscous fluid? Isn't that motion occurring at a much smaller Reynolds number so might be viscously controlling the phenomenon? Is this the estimate they are proposing for the Soret coefficient?*

Indeed, the particles are being displaced through a viscous fluid. This viscosity enters into the thermophoretic mobility of the particles through the diffusion coefficient D (eq. (1) of the main text), that is, it affects how fast an individual particle moves in a thermal gradient. However, experimentally, the timescale of the thermophoretic collapse is $100 \mu\text{s}$, which is much faster than what could be if the particle velocity was limited by viscous friction: the Reynolds number corresponding to the experimental situation is 10. This can be understood as during the collapse the particles move all at once in a spherical (in fact semi-spherical, which is the same save for boundary effects) geometry, hence the fluid experiences no shear.

Our estimate for the Soret coefficient, on the other hand, does not involve the thermophoretic collapse, but the slow formation of the particle aggregate once the force spiking mechanism has been jammed. The procedure for making our estimate is detailed in supple-

mentary figure S3.

(20) *Equation (6): μ_s is an odd choice for a density.*

We choose this notation because we refer to a mass density, while the letter ρ is already used for a number density.

(21) *I am puzzling over equation (9) – these estimates are based on a spherically symmetric collapse but then they are applied to the "explain" the order of magnitude of the force measured in the direction of motion. This seems inconsistent to me.*

We thank the referee for this relevant remark. Our reasoning is that a semi-spherical collapse, as occurs in reality, will give rise to the same fluid dynamics as a spherical collapse, save for boundary terms, which we may forget when dealing with orders of magnitude.

(22) *Did the authors ever explain the "Jeans instability"? I do not understand the analogy they are trying to make and possibly many readers will also not know the physics analogy being made?*

We agree with the referee in that the Jeans instability is not a generally known phenomenon and is worth explaining. **We have clarified the gravitational analogy in the revised manuscript**, in line with our answer to comment 15.

References

- [1] Versluis, M., Schmitz, B., Von der Heydt, A. & Lohse, D. How snapping shrimp snap: Through cavitating bubbles. *Science* **289**, 2114–2117 (2000).
- [2] Lauterborn, W. & Kurz, T. Physics of bubble oscillations. *Reports on Progress in Physics* **73**, 106501 (2010).
- [3] Borkent, B. M. *et al.* The acceleration of solid particles subjected to cavitation nucleation. *Journal of Fluid Mechanics* **610**, 157–182 (2008).
- [4] Hou, L., Yorulmaz, M., Verhart, N. R. & Orrit, M. Explosive formation and dynamics of vapor nanobubbles around a continuously heated gold nanosphere. *New Journal of Physics* **17**, 013050 (2015).
- [5] Jollans, T. & Orrit, M. Explosive, Oscillatory, and Leidenfrost Boiling at the Nanoscale (2019). [ArXiv1902.09331](https://arxiv.org/abs/1902.09331).
- [6] Wang, Y. *et al.* Giant and explosive plasmonic bubbles by delayed nucleation. *Proceedings of the National Academy of Sciences of the United States of America* **115**, 7676–7681 (2018).
- [7] Irimajiri, A., Hanai, T. & Inouye, A. A dielectric theory of "multi-stratified shell" model with its application to a lymphoma cell. *Journal of Theoretical Biology* **78**, 251–269 (1979).
- [8] Würger, A. Thermal non-equilibrium transport in colloids. *Reports on Progress in Physics* **73**, 126601 (2010).

- [9] Piazza, R. & Parola, A. Thermophoresis in colloidal suspensions. *Journal of Physics: Condensed Matter* **20**, 153102 (2008).
- [10] Braibanti, M., Vigolo, D. & Piazza, R. Does thermophoretic mobility depend on particle size? *Physical Review Letters* **100**, 1–4 (2008).
- [11] Duhr, S. & Braun, D. Why molecules move along a temperature gradient. *Proceedings of the National Academy of Sciences* **103**, 19678–19682 (2006).
- [12] Putnam, S. A. & Cahill, D. G. Transport of nanoscale latex spheres in a temperature gradient. *Langmuir* **21**, 5317–5323 (2005).
- [13] Helden, L., Eichhorn, R. & Bechinger, C. Direct measurement of thermophoretic forces. *Soft matter* **11**, 2379–2386 (2015).

REVIEWERS' COMMENTS:

Reviewer #2 (Remarks to the Author):

All questions have been nicely and convincingly answered. The paper is now ready for publication.

Reviewer #3 (Remarks to the Author):

The authors have done an outstanding job at revising their already good paper. Their additional edits and clarifications will help readers, and I anticipate that many will be interested in their paper. The paper is very well prepared and the figures are very well done to highlight the main observations and the mechanism, which I can better appreciate now. I strongly recommend publication.